# Isorhamnetin Induces Cell Cycle Arrest and Apoptosis Via Reactive Oxygen Species-Mediated AMP-Activated Protein Kinase Signaling Pathway Activation in Human Bladder Cancer Cells

**DOI:** 10.3390/cancers11101494

**Published:** 2019-10-04

**Authors:** Cheol Park, Hee-Jae Cha, Eun Ok Choi, Hyesook Lee, Hyun Hwang-Bo, Seon Yeong Ji, Min Yeong Kim, So Young Kim, Su Hyun Hong, JaeHun Cheong, Gi-Young Kim, Seok Joong Yun, Hye Jin Hwang, Wun-Jae Kim, Yung Hyun Choi

**Affiliations:** 1Department of Molecular Biology, College of Natural Sciences, Dong-eui University, Busan 47340, Korea; parkch@deu.ac.kr; 2Department of Parasitology and Genetics, Kosin University College of Medicine, Busan 49267, Korea; hcha@kosin.ac.kr; 3Anti-Aging Research Center, Dong-eui University, Busan 47227, Korea; nakajo39@naver.com (E.O.C.); 14769@deu.ac.kr (H.L.); hbhyun2003@naver.com (H.H.-B.); 14602@deu.ac.kr (S.Y.J.); ilytoo365@deu.ac.kr (M.Y.K.); 14731@deu.ac.kr (S.Y.K.); hongsh@deu.ac.kr (S.H.H.); 4Department of Biochemistry, Dong-eui University College of Korean Medicine, Busan 47227, Korea; 5Department of Molecular Biology, Pusan National University, Busan 46241, Korea; molecule85@pusan.ac.kr; 6Department of Marine Life Sciences, School of Marine Biomedical Sciences, Jeju National University, Jeju 63243, Korea; immunkim@jejunu.ac.kr; 7Department of Urology, College of Medicine, Chungbuk National University, Chungbuk 8644, Korea; sjyun@chungbuk.ac.kr; 8Department of Food and Nutrition, College of Nursing, Healthcare Sciences & Human Ecology, Dong-Eui University, Busan 47340, Korea; hhj2001@deu.ac.kr

**Keywords:** isorhamnetin, G2/M arrest, apoptosis, ROS, AMPK

## Abstract

Isorhamnetin is an O-methylated flavonol that is predominantly found in the fruits and leaves of various plants, which have been used for traditional herbal remedies. Although several previous studies have reported that this flavonol has diverse health-promoting effects, evidence is still lacking for the underlying molecular mechanism of its anti-cancer efficacy. In this study, we examined the anti-proliferative effect of isorhamnetin on human bladder cancer cells and found that isorhamnetin triggered the gap 2/ mitosis (G2/M) phase cell arrest and apoptosis. Our data showed that isorhamnetin decreased the expression of Wee1 and cyclin B1, but increased the expression of cyclin-dependent kinase (Cdk) inhibitor p21^WAF1/CIP1^, and increased p21 was bound to Cdk1. In addition, isorhamnetin-induced apoptosis was associated with the increased expression of the Fas/Fas ligand, reduced ratio of B-cell lymphoma 2 (Bcl-2)/Bcl-2 associated X protein (Bax) expression, cytosolic release of cytochrome *c*, and activation of caspases. Moreover, isorhamnetin inactivated the adenosine 5′-monophosphate-activated protein kinase (AMPK) signaling pathway by diminishing the adenosine triphosphate (ATP) production due to impaired mitochondrial function. Furthermore, isorhamnetin stimulated production of intracellular reactive oxygen species (ROS); however, the interruption of ROS generation using a ROS scavenger led to an escape from isorhamnetin-mediated G2/M arrest and apoptosis. Collectively, this is the first report to show that isorhamnetin inhibited the proliferation of human bladder cancer cells by ROS-dependent arrest of the cell cycle at the G2/M phase and induction of apoptosis. Therefore, our results provide an important basis for the interpretation of the anti-cancer mechanism of isorhamnetin in bladder cancer cells and support the rationale for the need to evaluate more precise molecular mechanisms and in vivo anti-cancer properties.

## 1. Introduction

Although new therapies for treating cancer patients are being developed, chemotherapy is still the main approach for the treatment of cancer. However, some limitations, such as adverse side effects, drug resistance, and limited efficacy, remain to be solved [1,2]. Therefore, urgent new therapeutic strategies that minimize these limitations and have high therapeutic efficacy are required. In this respect, there is an increasing interest in the importance of compounds derived from natural resources that have been traditionally used for the prevention and treatment of various diseases [3,4,5]. In particular, numerous naturally occurring agents have been reported to cause cell cycle arrest and induce apoptosis, which are important strategies for the control of proliferation in cancer cells, without inducing toxicity in normal cells [6,7]. These agents have also emerged as an alternative to chemopreventive and chemotherapeutic agents because they can specifically regulate various cellular signaling pathways in cancer cells [8,9]. 

Isorhamnetin (3′-methoxy-3,4′,5,7-tetrahydroxyflavone) is a flavonol aglycone found in some medicinal plants, such as *Hippophae rhamnoides* L., *Oenanthe javanica*, and *Ginkgo biloba* L., which are used as traditional medicines for the treatment of rheumatism, hemorrhage, cardiovascular disease, and cancer [10,11]. As one of the metabolites of quercetin, isorhamnetin is structurally similar to kaempferol, and is also called 3-O-methyl quercetin [12,13,14]. Isorhamnetin displays a number of biological properties due to its antioxidant, anti-inflammatory, and metabolic properties [15,16,17,18,19], and is also considered to have potential as an anti-cancer agent based on the results of various cancer cell models. For example, isorhamnetin has been reported to inhibit human leukemia, breast, colon, and cervical cancer cell proliferation through the gap 2/ mitosis (G2/M) phase arrest [20,21,22,23], and to induce mitotic block in non-small cell lung carcinoma cells, thus enhancing cisplatin- and carboplatin-induced G2/M arrest [24]. However, isorhamnetin induced S-phase arrest in some cancer cells [25,26], indicating that cell cycle arrest by isorhamnetin is dependent on the type of cancer cell line. 

In addition, the anti-cancer effects of isorhamnetin in various cancer cell lines have been shown to involve the death receptor (DR)-dependent extrinsic and/or mitochondria-dependent intrinsic pathways [19,24,27,28,29,30,31], which are representative apoptosis inducing pathways. It was also found that the anti-cancer effect of isorhamnetin was accompanied by the disturbance of various cellular signaling pathways [20,25,32]. Furthermore, isorhamnetin showed a strong cytotoxic effect through a reactive oxygen species (ROS)-dependent apoptosis pathway in breast cancer cells [26]. In particular, isorhamnetin was able to induce high cytotoxicity at low doses compared to quercetin in cancer cells, including hepatocellular carcinoma and leukemia cells [33,34]. Although the possibility of the growth inhibitory activity of isorhamnetin in bladder cancer cells has recently been proposed [35], no molecular mechanism has been reported to support its effect. Therefore, in this study, we investigated the anti-cancer efficacy of isorhamnetin in human bladder cancer cells, focusing on the mechanisms associated with the induction of cell cycle arrest and apoptosis.

## 2. Results

### 2.1. Isorhamnetin Inhibited Cell Viability in Bladder Cancer Cells

To examine the cytotoxic effect of isorhamnetin, four bladder cancer T24 cell lines (T24, 5637, and 2531J) were treated with various concentrations of isorhamnetin, and then the 3-(4,5-dimethyl-2-thiazolyl)-2,5-diphenyltetra-zolium bromide (MTT) assay was conducted. Although there are some differences depending on the cell line, the cell viability was significantly decreased in a concentration-dependent manner in isorhamnetin-treated cells (Figure 1A), without affecting normal cultured human keratinocyte HaCaT cells and Chang liver cells under the same conditions. In addition, the 50% inhibitory concentration (IC_50_) values of isorhamnetin on T24 and 5637 cells were 127.86 μM and 145.75 μM, respectively. The microscopic examination demonstrated that the phenotypic characteristics of isorhamnetin-treated T24 and 5637 cells showed irregular cell outlines, a decrease of cell density, shrinkage, and an increase of detached cells (Figure 1B, upper panel). In addition, 2531J cells showed similar results from the isorhamnetin treatment.

### 2.2. Isorhamnetin Induced G2/M Phase Arrest and Apoptosis in Bladder Cancer Cells

To examine the mechanism responsible for the isorhamnetin-induced anti-proliferative effect, the cell cycle distribution profile was examined. Flow cytometry data demonstrated that the percentage of cells arrested at G2/M phase was increased with increasing isorhamnetin treatment concentration, coupled with a decrease in the proportion of cells in the G1 and S phases (Figure 1C). Meanwhile, a significant increase of the cells in the sub-G1 phase, which was used as an index of apoptotic cells, was observed in isorhamnetin-treated cells (Figure 1D). Therefore, 4′,6-diamidino-2-phenylindole (DAPI) staining was performed to investigate whether apoptosis was involved in cell the growth inhibition induced by isorhamnetin. Figure 1B (lower panel) indicates that morphological changes of the nuclei, which were observed in apoptosis-inducing cells, such as nuclear fragmentation and chromatin condensation, were dominantly found in isorhamnetin-treated T24 and 5637 cells. Since 2531J cells also had the same results, the following experiments were performed on T24 and 5637 cells. To quantify the apoptosis triggered by isorhamnetin, an annexin V-fluorescein isothiocyanate (FITC)/propidium iodide (PI) double staining assay was conducted. As indicated in Figure 2A,B, after treatment with isorhamnetin, the populations of annexin V-staining positive cells were significantly increased, as compared to the control. On the other hand, T24 cells showed slightly increase in necrotic death upon 100 μM of isorhamnetin, but not 5637 cells (Figure 2B). Consistent with this, the results from agarose gel electrophoresis showed that as the isorhamnetin concentration increased and more fragmented DNA was observed (Figure 2C), indicating that an isorhamnetin-induced G2/M phase arrest was associated with the induction of apoptosis. 

### 2.3. Isorhamnetin Regulated the Expression of G2/M Phase-Associated Proteins in Bladder Cancer Cells

To explore the biochemical event of the isorhamnetin-elicited cell cycle arrest, the levels of G2/M phase-associated proteins were analyzed. The immunoblotting results revealed that following isorhamnetin treatment, the levels of Wee1 and cyclin B1 were reduced, and the effect was concentration-dependent, while the expression of cyclin-dependent kinase (Cdk) 1 (also called cell division cycle 2, Cdc2) was maintained at the level of the control group (Figure 3A,B). However, the expression of Cdk inhibitor p21^WAF1/CIP1^ was markedly increased in response to isorhamnetin exposure. Next, we performed co-immunoprecipitation to investigate the role of isorhamnetin-induced p21, and found that this increased p21 via treatment with isorhamnetin complexed with Cdk1 (Figure 3C). These results suggest that increased p21 protein in isorhamnetin-treated cells contributed to G2/M phase arrest by inhibiting its activity through binding to Cdk1.

### 2.4. Isorhamnetin Modulated the Expression of Apoptosis-Regulatory Proteins in Bladder Cancer Cells

To investigate the pathway of isorhamnetin-induced apoptosis, caspases activities were determined. Figure 4A,B shows that the protein levels of pro-caspase-8, -9, and -3 were concentration-dependently decreased, which was associated with the degradation of poly(ADP-ribose) polymerase (PARP). Therefore, we quantitatively assessed each caspase activity in the presence of isorhamnetin using fluorogenic substrates to determine whether these immunoblotting results were directly related to activation of the corresponding caspases and found that treatment with isorhamnetin significantly stimulated the activation of these caspases in a concentration-dependent manner in comparison with untreated control cells (Figure 4C). In addition, the effects of isorhamnetin on the expression of the Fas/Fas ligand (FasL) and B-cell lymphoma 2 (Bcl-2) family members were determined. Figure 4A,B shows that both Fas and FasL protein levels were up-regulated, and the level of Bcl-2-associated X protein (Bax), a pro-apoptotic protein, was also increased, while the level of Bcl-2, an anti-apoptotic protein, was reduced in isorhamnetin-treated cells. Furthermore, isorhamnetin promoted the release of cytochrome *c* from mitochondria into cytosol (Figure 4D).

### 2.5. Isorhamnetin Increased ROS Generation but Decreased ATP Content in Cancer Cells

To investigate the involvement of ROS on the cytotoxic effect of isorhamnetin, we performed flow cytometry analysis using a fluorescent probe, 5,6-carboxy-2′,7′-dichlorodihydrofluorescein diacetate (DCF-DA). Our data indicated that the production of ROS showed a significant increase within 1 h of the isorhamnetin treatment, and then gradually decreased, while the antioxidant N-acetyl-L-cysteine (NAC) suppressed it to the control level (Figure 5A–C). In addition, an ATP colorimetric assay kit was used to measure the content of mitochondrial ATP in the cells. Figure 5D shows that the concentration of ATP in the isorhamnetin-treated cells decreased in a concentration-dependent manner. However, under the condition that NAC existed, it was markedly weakened, indicating that the decrease in ATP levels was associated with ROS production.

### 2.6. Isorhamnetin Reduced Mitochondrial Membrane Potential (MMP, ΔΨm) and Activated Adenosine 5’-Monophosphate-Activated Protein Kinase (AMPK) Signaling in Bladder Cancer Cells

We assessed the level of MMP to investigate whether the inhibition of ROS-dependent ATP production by isorhamnetin was associated with impaired mitochondrial function. According to the results of flow cytometry using 5,5′,6,6′-tetrachloro-1,1′,3,3′-tetraethylimidacarbocyanine iodide (JC-1) dyes, the formation of JC-1 aggregates in mitochondria was maintained at a relatively high rate in cells not treated with isorhamnetin, while the ratio of JC-1 monomers increased with increasing isorhamnetin treatment concentration, indicating a remarkable depletion of MMP after isorhamnetin treatment (Figure 6A,B). Furthermore, isorhamnetin increased the phosphorylated level of AMPK, as well as its downstream factor acetyl-CoA carboxylase (ACC) (Figure 6C,F), indicating that the AMPK signaling pathway was activated as a result of the loss of ATP. Additionally, we evaluated the effects of isorhamnetin on the phosphorylation of the mechanistic target of rapamycin (mTOR), p70S6K, and Unc-51-like kinase (ULK1), which are AMPK downstream molecules that regulate cell proliferation, apoptosis, and autophagy [36]. Exposure of T24 and 5637 cells with isorhamnetin led to down-regulation in the phosphorylation of mTOR and p76S6K in a dose-dependent manner (Figure 6F,G). Interestingly, we found the isorhamnetin inhibited autophagy via down-regulation of the expression and phosphorylation of ULK1. In addition, our supplementary result showed that the expression of autophagy-related markers was down-regulated using the isorhamnetin treatment, similar to ULK1 (Appendix A). However, the presence of NAC or compound C, an antagonist of AMPK, significantly prevented the isorhamnetin-induced loss of MMP (Figure 6A,B), and NAC or compound C also markedly abolished enhanced activation of the AMPK signaling by isorhamnetin (Figure 6E). These data indicate that isorhamnetin-promoted mitochondrial dysfunction associated with the disturbance of ATP production was mediated through an ROS-dependent pathway.

### 2.7. ROS Acted as an Upstream Regulator of Isorhamnetin-Mediated Apoptosis and Cell Cycle Blockade in Bladder Cancer Cells

The effect of ROS on isorhamnetin-mediated apoptosis and G2/M phase arrest was further investigated to determine the role of ROS in the anti-cancer activity of isorhamnetin. As depicted in the results of the DAPI staining and flow cytometry analysis, artificially blocking the production of ROS using NAC drastically attenuated isorhamnetin-induced apoptosis (Figure 7A–C). In parallel, pretreatment with NAC protected isorhamnetin-mediated G2/M arrest, which was related to a decrease in the number of sub-G1 phase cells (Figure 7E). Consistent with these results, inhibiting ROS production greatly restored reduced cell viability using isorhamnetin (Figure 7F), demonstrating that ROS generation was shown to be necessary for the contribution of apoptosis and G2/M arrest using isorhamnetin.

## 3. Discussion

In many previous studies, it is clear that the induction of apoptosis by many anti-cancer agents is associated with cell cycle arrest at specific checkpoints [6,37]. In particular, the deregulation of cell cycle control is clearly implicated in the development and progression of most tumors, and the interruption of this progression is considered to be an important strategy to inhibit the proliferation of cancer cells [6,37]. Therefore, we first investigated whether the suppression of bladder cancer cell proliferation by isorhamnetin was associated with cell cycle arrest. The results of flow cytometry analysis showed that isorhamnetin caused G2/M phase arrest, similar to the results of previous studies in several human cancer cell lines [20,21,22,23], suggesting that G2/M phase arrest is one of the mechanisms of the growth inhibitory effects of isorhamnetin in human bladder cancer cells. The progression of the cell cycle in eukaryotic cells is tightly controlled by the interaction of cyclins and Cdks with their inhibitory factors [38,39]. In this process, the transition from G2 to M phase is achieved through the increased activity of Cdk1 by cyclin B1 complexing with Cdk1. In addition, Wee1 is a tyrosine kinase that induces phosphorylation of Cdk1, resulting in inhibition of cyclin B-Cdk1 activity and preventing cell mitotic entry [40,41]. In the current study, exposure of bladder cancer cells to isorhamnetin markedly reduced the expression of cyclin B1 and Wee1, without significant changes in the expression of Cdk1. 

p21, a typical Cdk inhibitor belonging to the kinase inhibitory protein/ CDK interacting protein (KIP/CIP) family, has a broad-spectrum of specificity in the cell cycle proteins [38,39]. p21 was first reported to be a major inducer of tumor suppressor p53-dependent cell cycle arrest induced by DNA damage, but it could act as a mediator of p53-independent cell arrest in various types of cancer cells [42,43]. As a Cdk inhibitor, when p21 expression increases, it forms complexes with Cdks, reducing their kinase activity and inhibiting cell cycle progression [42,44]. According to our data, isorhamnetin dramatically increased p21 levels with increasing treatment concentration, and increased p21 complexed with Cdk1, which might have contributed to the inhibition of Cdk1 kinase activity. In addition, since T24 and 5637 cells are mutant p53 gene-bearing cell lines [45], increased p21 expression using isorhamnetin was thought to contribute to G2/M arrest, regardless of p53 gene status. Collectively, our data suggest that isorhamnetin-triggered G2/M arrest was due to the decreased expression of Wee1 and cyclin B1, and inactivation of p53-independent p21-mediated Cdk1 kinase.

Because the induction of apoptosis in cancer cells along with cell cycle arrest is a promising approach to cancer therapy, we assessed whether G2/M arrest using isorhamnetin was associated with apoptosis induction. Based on the results of morphological changes, DNA fragmentation, and flow cytometry analysis, we found that the cytotoxic effect of isorhamnetin was achieved through the induction of apoptosis associated with G2/M arrest. As is well known, apoptosis can be largely categorized into extrinsic and intrinsic pathways in mammalian cells [37,46]. The extrinsic pathway is characterized by the activation of caspase-8 by the formation of the death-inducing signal complex through the binding of death ligands to the cell surface DRs. For example, when FasL, one of the typical death ligands, binds to the corresponding DR, Fas, caspase-8 is sequentially activated [46,47]. On the other hand, the intrinsic pathway begins via the activation of caspase-9 through the release of mitochondrial pro-apoptotic proteins, such as cytochrome *c*, from mitochondria to cytoplasm due to increased mitochondrial permeability. This pathway is tightly regulated by the Bcl-2 protein family that includes pro- and anti-apoptotic proteins, which guard mitochondrial integrity and control the release of cytochrome *c* through the mitochondrial transition pore [48,49]. Caspases-8 and -9, which correspond to the initiator caspases of each pathway, ultimately activate apoptosis through the cleavage of various cellular substrates, such as PARP, by activating downstream executioner caspases, including caspase-3 and -7 [37,50]. In addition, these pathways are strictly controlled by a variety of cellular signaling pathways and regulatory molecules [50,51]. Our results show that isorhamnetin increased the expression of Fas and FasL; activated caspase-8, -9, and -3; and induced the cleavage of PARP. In addition, consistent with previous studies in non-small cell lung cancer cells and Lewis lung cancer cells [31,52], mitochondrial dysfunction was induced, as confirmed by the loss of MMP in isorhamnetin-treated cells. Moreover, the loss of MMP was accompanied by a down-regulation in the Bcl-2/Bax ratio and the promotion of cytosolic release of cytochrome *c*. Therefore, based on those observations, we speculated that the pro-apoptotic effect of isorhamnetin in bladder cancer cells could occur by simultaneously activating extrinsic and intrinsic pathways. 

Growing evidence demonstrates that many anti-cancer agents induce apoptosis through pro-oxidant properties, such as increasing ROS accumulation or destroying cellular antioxidant systems [8,53]. In particular, mitochondria are the major subcellular organelles responsible for the production of ROS in the cells and are also a major target of ROS [54,55]. Therefore, elevating intracellular levels of ROS production is considered to be one of the ideal mechanisms for killing cancer cells through the activation of intrinsic pathways. Intriguingly, in various cell types, ROS are involved in activating the signaling system of AMPK, a key sensor that regulates energy balance and cell fate [56,57,58]. Mitochondrial dysfunction, due to excessive production of ROS, leads to a loss of function of the respiratory chain in the mitochondrial inner membrane, which can lower intracellular ATP levels and activate AMPK [56,57]. Choi et al. first reported that ROS induces concentration-dependent activation of AMPK [59]. More recently, it has been described that AMPK can be activated by ROS, thereby leading to an increase of glycolysis [60,61]. Furthermore, Corton et al. reported that hypoxic activation of AMPK was dependent on the levels of the mitochondrial ROS [62], and Tavazzi et al. demonstrated that AMPK activation was caused by ROS-mediated intracellular ATP depletion [63]. On the contrary, it has been reported that treatment of the potent ROS scavengers, including NAC and dimethyl sulfoxide (DMSO), significantly abolished oxidative stress-induced AMPK activation and ATP depletion [60,61,64]. Consistent with a previous study in breast cancer cells [22], our results show that isorhamnetin treatment markedly increased the levels of ROS production; however, the ROS scavenger, NAC, greatly weakened the accumulation of ROS by isorhamnetin. The quenching of ROS generation also significantly diminished the isorhamnetin-induced disruption of MMP to the control level, followed by significant ATP restoration, indicating that ROS act as upstream signaling molecules to enhance isorhamnetin-mediated mitochondrial dysfunction. Our results also demonstrate that the activation of the AMPK signaling pathway was increased in cells exposed to isorhamnetin, probably due to decreased ATP content. Furthermore, the presence of NAC markedly attenuated isorhamnetin-induced phosphorylation of AMPK, while their total protein levels were kept at an equivalent level, suggesting that the isorhamnetin-induced activation of AMPK signaling pathway is dependent on ROS production. Subsequently, NAC pretreatment also significantly reversed the enhanced apoptosis, G2/M phase arrest, and viability reduction induced by isorhamnetin, confirming that increasing ROS may serve as a key contributor to the anti-cancer effects of isorhamnetin. The AMPK acts as a metabolic mater switch that controls cell fate, such as cell survival, apoptosis, and autophagy [65]. Indeed, fatty acid synthesis is a critical energy-consuming process for the differentiation of tumor cells, and it has been demonstrated that AMPK inhibits lipid synthesis by the phosphorylation and inactivation of acetyl-CoA carboxylase 1 (ACC1) [65]. Furthermore, AMPK directly inhibits mTOR complex I, which regulates p70S6K, an enhancer of protein synthesis. In this sense, AMPK plays a critical role as a cell growth suppressor by inhibiting protein, rRNA, and lipid synthesis [65,66]. In the present study, we conjecture that the AMPK-mediated interruption of the mTOR/p70S6K/ACC1 signaling pathway may contribute to isorhamnetin-induced cell cycle arrest and apoptosis. On the other hand, there are conflicting opinions on the relationship between AMPK and autophagy. Although increasing evidence described that AMPK activation can induce the autophagy through the inhibition of mTOR and phosphorylation of ULK1 [67,68], a few studies reported that ROS attenuated autophagy by the down-regulation of ULK-1 [69,70]. Interestingly, based on our results, we found the isorhamnetin inhibited autophagy by down-regulation of the expression and phosphorylation of ULK1. Therefore, our data suggested that isorhamnetin-induced ROS activates AMPK, and subsequently down-regulates the mTOR/ACC1/ULK1 signaling pathway, which results in promoting cell apoptosis and inhibits autophagy at the same time.

The current results lead us to suggest that the production of ROS by isorhamnetin plays a critical role in the induction of G2/M arrest and apoptosis through simultaneous initiation of both extrinsic and intrinsic pathways in human bladder cancer cells. In addition, ROS act as an upstream signal related to the effect of isorhamnetin on the activation of the AMPK signaling pathway. However, further studies are warranted to identify the molecular mechanisms of isorhamnetin-mediated activation of AMPK signaling on autophagy and mitochondrial energy metabolism in bladder cancer cells. In addition, further studies are required to identify and understand the role of intracellular organelles involved in ROS generation by isorhamnetin, including in vivo animal experiments.

## 4. Materials and Methods 

### 4.1. Cell Culture and Isorhamnetin Treatment 

The human bladder cancer cell lines (T24, 5637, 2531J, and EJ) were purchased from the American Type Culture Collection (Manassas, MD, USA). The cells were cultured in Dulbecco’s modified Eagle’s medium supplemented with 10% fetal bovine serum and 1% penicillin-streptomycin (100 U/mL penicillin and 100 μg/mL streptomycin, all from WelGENE Inc., Daegu, Republic of Korea) at 37 °C under a humidified 5% CO_2_. The cells were sub-cultured every 3–4 days to maintain logarithmic growth, and were allowed to grow for 24 h before treatments were applied. Isorhamnetin was obtained from Sigma-Aldrich Chemical Co. (St. Louis, MO, USA), and was dissolved in dimethyl sulfoxide (DMSO, Sigma-Aldrich Chemical Co.) to a final concentration of 100 M. Prior to use, the stock solution was diluted with culture medium to the desired concentration. 

### 4.2. Cell Viability Assay 

Cell viability was determined using an MTT assay, as previously described [71]. Briefly, cells (1 × 10^4^ cells/well) were seeded onto 96-well plates in 100 μL medium. After overnight incubation, the cells were exposed to a series of concentrations of isorhamnetin for 48 h. Thereafter, the MTT reagent (Sigma-Aldrich Chemical Co.) at 50 μg/mL final concentration was added to each well and cells were incubated continuously at 37 °C for 2 h. The medium was then removed and 100 μL DMSO was added to each well to dissolve the formed blue formazan crystals, followed by measurement at 540 nm in a microplate reader (Molecular Device Co., Sunnyvale, CA, USA). All results were performed in three independent experiments and the cell survival rate was expressed as a percentage of the control. The morphological changes of cells were directly observed and photographed using phase-contrast microscopy (Carl Zeiss, Oberkochen, Germany). 

### 4.3. Determination of Cell Cycle Distribution Using Flow Cytometric Analysis 

PI staining was applied to analyze the DNA content and cell cycle distribution. In brief, cells were exposed to different concentrations of isorhamnetin for 48 h, and then the cells were harvested and fixed gently in 70% ice-cold ethanol (in phosphate-buffered saline, PBS, WelGENE Inc., Daegu, Korea) at 4 °C for 30 min. The cells were re-suspended in PBS containing 40 μg/mL PI, 100 μg/mL RNase A, and 0.1% triton X-100 (all from Sigma-Aldrich Chemical Co.) in a dark room at 37 °C for 30 min, and subjected to flow cytometry (BD Biosciences, San Jose, CA, USA), to determine the cell cycle distribution and apoptotic cells (sub-G1 phase). 

### 4.4. Determination of Apoptotic Cell Death by Flow Cytometric Analysis 

The Annexin V-FITC staining kit from BD Biosciences (San Jose, CA, USA) was used to determine and quantify the apoptotic cells using flow cytometry, according to the manufacturer’s instruction. In brief, the collected cells were suspended in the supplied binding buffer, and then stained with FITC-conjugated annexin V and PI at room temperature (RT) for 20 min in the dark. The fluorescent intensities of the cells were detected using flow cytometry, and the annexin V^+^/PI^−^ and annexin V^+^/PI^+^ cell populations were considered indicators of apoptotic cells. 

### 4.5. Nuclear Staining and Deoxyribonucleic Acid (DAN) Fragmentation Assay 

The changes of nuclear morphology for assessing apoptosis were assessed using DAPI staining. Briefly, cells were cultured with or without different concentrations of isorhamnetin for 48 h, and were then fixed with 4% paraformaldehyde (Sigma-Aldrich Chemical Co.) for 10 min at RT. The cells were rinsed with PBS, and incubated with 1 μg/mL DAPI solution (Sigma-Aldrich Chemical Co.) at 37 °C for 10 min. Stained cells were visualized and photographed using fluorescence microscopy (Carl Zeiss, Oberkochen, Germany). For DNA fragmentation assay, the collected cells were lysed in a buffer containing 10 mM Tris-HCl (pH 7.4), 150 mM NaCl, 5 mM ethylenediaminetetraacetic acid, and 0.5% Triton X-100 for 30 min. The fragmented DNA in the supernatant was extracted using an equal volume of neutral phenol:chloroform:isoamyl alcohol (25:24:1, Sigma-Aldrich Chemical Co.), analyzed electrophoretically on 1% agarose gel containing EtBr (Sigma-Aldrich Chemical Co.), and photographed under a Fusion FX Image system (Vilber Lourmat, Torcy, France).

### 4.6. Protein Extraction, Co-Immunoprecipitation, and Western Blot Analysis 

After treatment, both adherent and floating cells were harvested, and the whole cellular proteins were prepared using the Bradford protein assay kit (Bio-Rad Laboratories, Hercules, CA, USA), according to the manufacturer’s protocol. For the preparation of mitochondrial and cytosolic proteins from the cells, NE-PER nuclear and cytoplasmic extraction reagents (Thermo Fisher Scientific Inc., Waltham, UT, USA) were applied. Protein concentration was measured using the Bio-Rad protein assay kit (Bio-Rad Laboratories, Hercules, CA, USA), according to the manufacturer’s instructions. For the co-immunoprecipitation assay, the 500 μg of cell lysates from each sample was precleaned with normal rabbit immunoglobulin G (IgG) and a protein-A-sepharose bead slurry (Amersham, Arlington Heights, IL, USA), and immunoprecipitation was conducted using 1 μg of anti-Cdk1 antibody (Santa Cruz Biotechnology, Inc., Santa Cruz, CA, USA) and protein-A-sepharose (Sigma-Aldrich Chemical Co.). The protein complex was then prepared according to the previously described method [62]. For Western blot analysis, equal amounts of protein samples or immunoprecipitated proteins were separated using sodium dodecyl sulphate-polyacrylamide gel electrophoresis and transferred to polyvinylidene difluoride membranes (Millipore, Bedford, MA, USA) (whole blot figures can be found at the Appendix A). The membranes were blocked with Tris-buffered saline (10 mM Tris-Cl, pH 7.4) containing 0.5% Tween-20 and 5% nonfat dry milk for 1 h at RT, and then probed with the indicated primary antibodies (Santa Cruz Biotechnology, Inc., and Cell Signaling Technology, Danvers, MA, USA), to react with the blotted membranes at 4 °C overnight. Afterwards, the membranes were incubated with the corresponding horseradish peroxidase-conjugated secondary antibodies (Santa Cruz Biotechnology, Inc.), developed using an ECL detection kit (GE Healthcare Life Sciences, Little Chalfont, U.K.), and then visualized using a Fusion FX Image system. Densitometric analysis of the data was performed using the ImageJ^®^ software (v1.48, NIH, Bethesda, MD, USA).

### 4.7. Caspase Activity Assay 

The activity of caspases was measured according to the manufacturer’s instructions for the Caspase colorimetric assay kits (R&D Systems, Minneapolis, MN, USA). Briefly, cells were harvested and lysed in the lysis buffer provided in the kit on ice for 10 min, and then centrifuged at 10,000× *g* for 1 min. The supernatants containing equal proteins were incubated with the supplied reaction mixtures, including the fluorogenic peptide substrate (Asp-Glu-Val-Asp specific for caspase-3, Ile-Glu-Thr-Asp for caspase-8, and Leu-Glu-His-Asp specific for caspase-9) labeled with p-nitroaniline (pNA) for 1 h at 37 °C in the dark. The amounts of released pNA was measured using a microplate reader using excitement at 405 nm and emitting at 510 nm. 

### 4.8. Measurement of ROS Production and MMP

The production of ROS was measured using DCF-DA, as described previously [72]. At the end of the treatment with isorhamnetin for defined periods in the presence or absence of NAC (Sigma-Aldrich Chemical Co.), cells were washed with PBS and incubated with 10 μM DCF-DA (Invitrogen, Carlsbad, CA, USA) in the dark at 37 °C for 20 min. Subsequently, cells were analyzed for DCF fluorescence using flow cytometry at 480 nm/520 nm. To measure MMP, JC-1 staining was performed according to the manufacturer’s instructions. After treatment with isorhamnetin for 48 h in the presence or absence of NAC or compound C, cells were exposed to 10 μM JC-1 (Invitrogen) for 30 min at 37 °C, and then analyzed using flow cytometry at 488 nm/575 nm, as previously described [73]. 

### 4.9. Detection of ATP Levels 

The firefly luciferase-based ATP Bioluminescence assay kit (Roche Applied Science, Indianapolis, IN, USA) was used for the detection of intracellular ATP levels, according to the manufacturer’s instructions. Briefly, cells treated with isorhamnetin for 48 h with or without NAC were lysed with the lysis buffer provided in the kit, and the supernatants were collected via centrifugation at 12,000× *g* for 10 min at 4 °C. Subsequently, an equal amount of supernatants and ATP detection reagent, which catalyzed the light production from ATP and luciferin, were mixed. Firefly luciferase activity was immediately measured using a luminometer and the ATP level was calculated according to the ATP standard curve. Intracellular ATP levels were calculated as a percentage of the untreated control.

### 4.10. Statistical Analysis 

All experiments were performed at least three times. Data were analyzed using GraphPad Prism software (version 5.03; GraphPad Software, Inc., La Jolla, CA, USA), and expressed as the mean ± standard deviation (SD). Differences between groups were assessed using analysis of variance, followed by ANOVA-Tukey’s post hoc test, and *p* < 0.05 was considered to indicate a statistically significant difference. 

## 5. Conclusions

Our findings demonstrate that isorhamnetin exerted an anti-proliferative effect on human bladder cancer cells through the induction of cell cycle arrest during the G2/M phase and apoptosis. Isorhamnetin-induced G2/M arrest was attributed to the decrease in Wee1 and cyclin B1 expression and the upregulation of p21. Isorhamnetin also induced apoptosis by activating caspase-8 and -9, which belong to the initiator caspases of the extrinsic and intrinsic pathways, respectively, followed by the activation of effector caspase-3, leading to the degradation of PARP. In addition, isorhamnetin enhanced the mitochondrial dysfunction, which was associated with an increase in Bax/Bcl-2 expression ratio and cytochrome *c* release into the cytoplasm. Moreover, the induction of G2/M arrest and apoptosis by isorhamnetin was accompanied by activation of the AMPK signaling pathway, and excessive production of ROS. However, artificial interception of the AMPK signaling pathway attenuated isorhamnetin-induced apoptosis, and the interruption of ROS generation led cells to escape from G2/M arrest and apoptosis. Based on these finding, we suggest that isorhamnetin has chemopreventive potential by inducing G2/M arrest and apoptosis through ROS-dependent activation of the AMPK signaling pathway in bladder cancer cells.

## Figures and Tables

**Figure 1 cancers-11-01494-f001:**
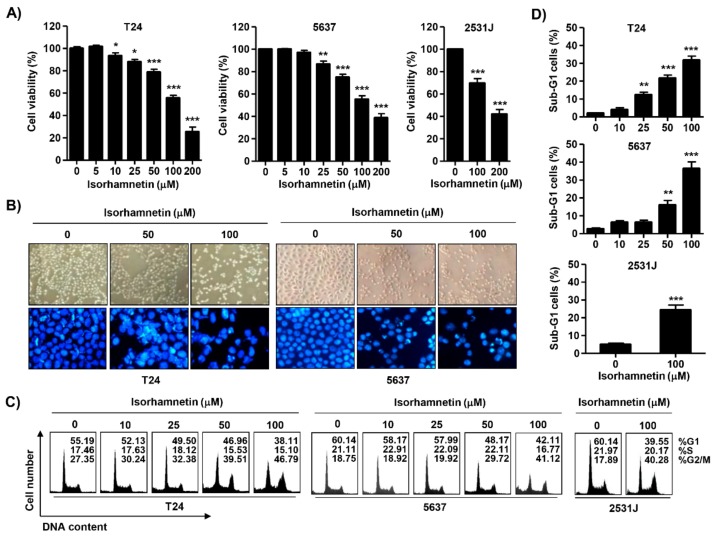
The inhibition of cell viability and induction of cell cycle arrest at gap 2/ mitosis (G2/M) phase using isorhamnetin in bladder cancer cells. T24, 5637, and 2531J cells were treated with the indicated concentrations of isorhamnetin for 48 h. (**A**) The cell viability was assessed using 3-(4,5-dimethyl-2-thiazolyl)-2,5-diphenyltetra-zolium bromide (MTT) assay. Each bar represents the mean ± standard deviation (SD) of three independent experiments (* *p* < 0.05 and *** *p* < 0.0001 compared to the control). (**B**, Upper panel) Morphological changes of T24 and 5637 cells were observed using phase-contrast microscopy. (B, Lower panel) The 4′,6-diamidino-2-phenylindole (DAPI)-stained nuclei were pictured under a fluorescence microscope. Representative photographs of the morphological changes are presented. (**C**,**D**) The cells were stained with propidium iodide (PI) solution for flow cytometry analysis. Quantification of the cell population (in percent) in different cell cycle phases of viable cells is shown. (D) Sub-G 1% was calculated as the percentage of the number of cells in the sub-G1 population relative to the number of total cells. Data were expressed as the mean ± SD of three independent experiments (* *p* < 0.05, ** *p* < 0.001, and *** *p* < 0.0001 compared to the control).

**Figure 2 cancers-11-01494-f002:**
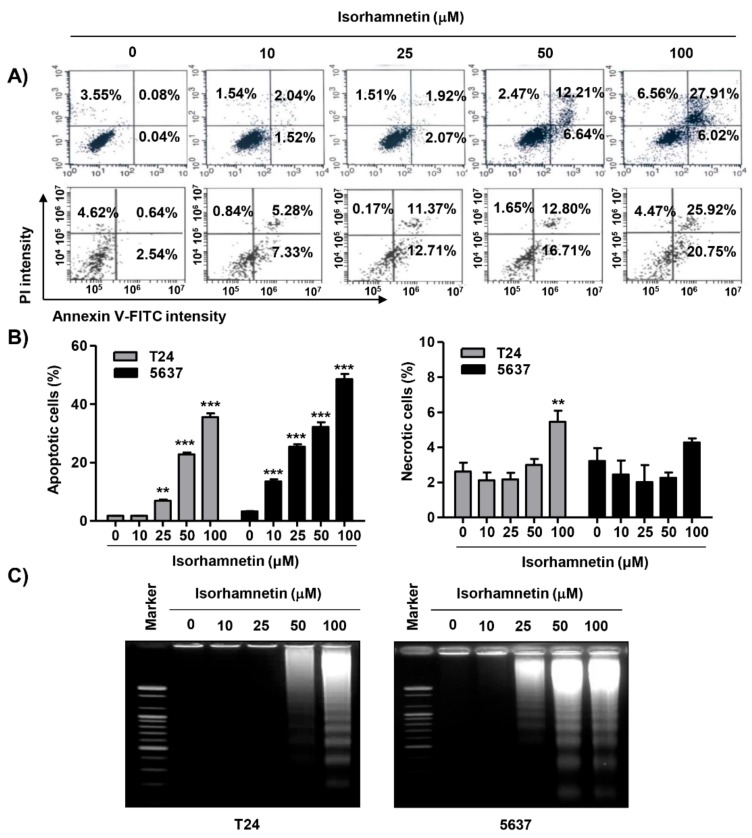
Induction of apoptosis using isorhamnetin in bladder cancer cells. (**A**,**B**) After treatment with different concentrations of isorhamnetin for 48 h, the cells were fixed and stained with annexin V-fluorescein isothiocyanate (FITC) and PI for flow cytometry analysis. (A) Representative profiles. The results show early apoptosis, defined as annexin V^+^ and PI^−^ cells (lower right quadrant), and late apoptosis, defined as annexin V^+^ and PI^+^ (upper right quadrant) cells. (**B**) The percentages of apoptotic cells (left) and necrotic cells (right) were determined by expressing the numbers of Annexin V^+^ cells as percentages of all the present cells. The results are presented as the mean ± SD of three independent experiments (** *p* < 0.001 and *** *p* < 0.0001 compared to the control). (**C**) DNA fragmentation in the cells cultured under the same conditions was analyzed via the extraction of genomic DNA, electrophoresis in agarose gel, and then visualization using ethidium bromide (EtBr) staining.

**Figure 3 cancers-11-01494-f003:**
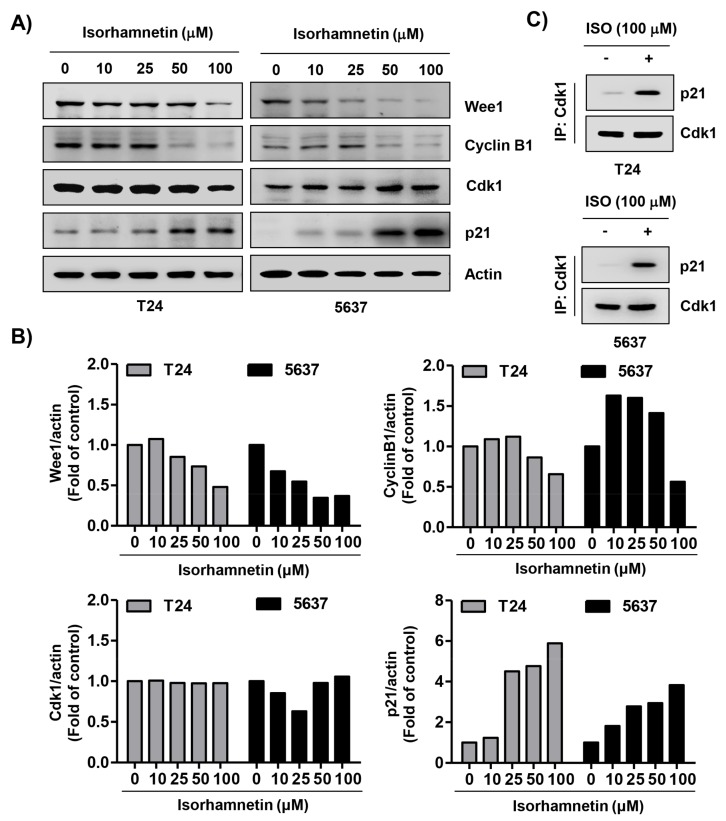
Effects of isorhamnetin on the levels of cell cycle regulatory proteins in bladder cancer cells. (**A**) T24 and 5637 cells were treated with the indicated concentrations of isorhamnetin for 48 h, and then total cell lysates were prepared. Western blotting was then performed using the indicated antibodies and an enhanced chemiluminescence (ECL) detection system. Actin was used as an internal control. (**B**) The expression of each protein was indicated as a fold change relative to the control. Quantitative analysis of mean pixel density was performed using the ImageJ^®^ software. (**C**) Cells were incubated without or with 100 μM isorhamnetin for 48 h, and then equal amounts of proteins were immunoprecipitated with the anti-cyclin-dependent kinase (Cdk) 1 antibody. Western blotting using immunocomplexes was performed using anti-p21 or anti-Cdk1 antibodies and an ECL detection system (IP, immunoprecipitation).

**Figure 4 cancers-11-01494-f004:**
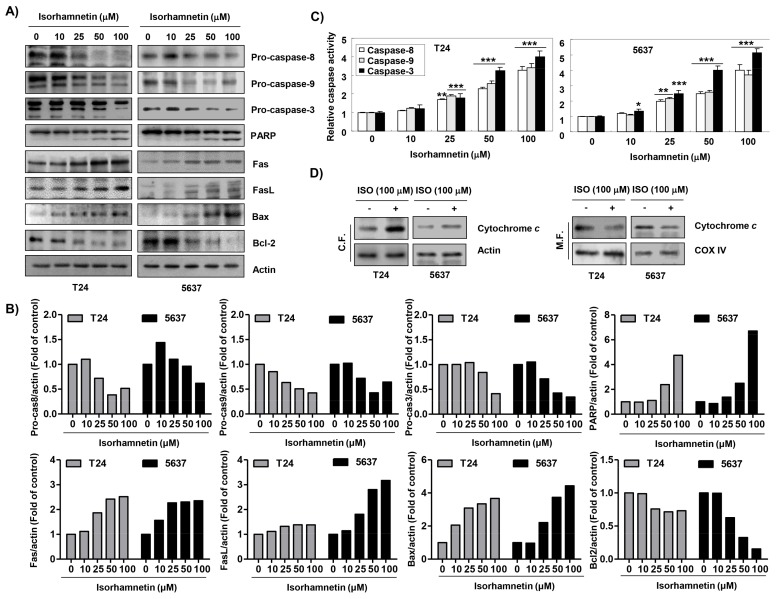
Modulation of apoptosis-regulatory factors using isorhamnetin in bladder cancer cells. (**A**) After treatment with isorhamnetin for 48 h, Western blotting was performed using the indicated antibodies and an ECL detection system. Actin was used as an internal control. (**B**) The expression of each protein was indicated as a fold change relative to the control. Quantitative analysis of mean pixel density was performed using the ImageJ^®^ software. (**C**) The activities of caspases were evaluated using caspase colorimetric assay kits. The data were expressed as the mean ± SD of three independent experiments (* *p* < 0.05, ** *p* < 0.001, and *** *p* < 0.0001 compared to the control). (**D**) After treatment without or with 100 μM isorhamnetin for 48 h, cytosolic and mitochondrial proteins were prepared and analyzed for cytochrome *c* expression using Western blot analysis. Equal protein loading was confirmed via the analysis of actin and cytochrome oxidase subunit VI (COX VI) in each protein extract (C.F.—cytosolic fraction; M.F.—mitochondrial fraction).

**Figure 5 cancers-11-01494-f005:**
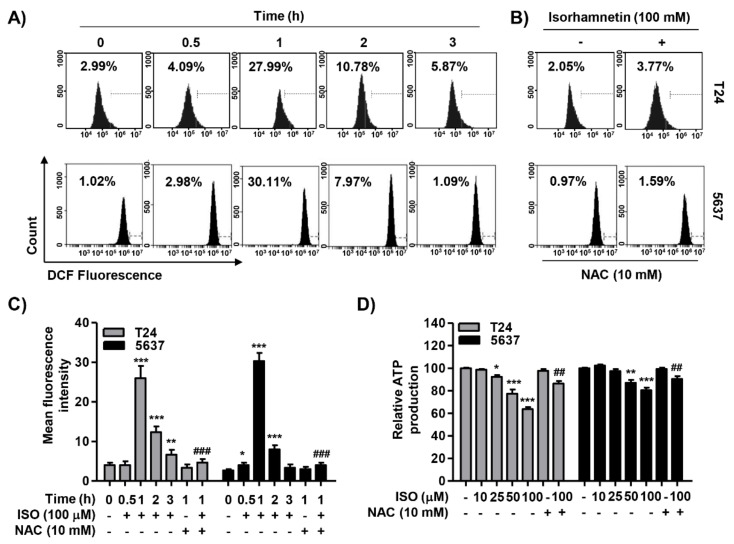
Accumulation of reactive oxygen species (ROS) and decrease of ATP content using isorhamnetin in bladder cancer cells. (**A**) Cells were treated with 100 μM isorhamnetin for the indicated times. (**B**) The cells were pre-treated with or without 10 mM N-acetyl-L-cysteine (NAC) for 1 h before isorhamnetin treatment for 1 h. (A,B) The medium was discarded and the cells were incubated for 20 min with medium containing 5,6-carboxy-2′,7′-dichlorodihydrofluorescein diacetate (DCF-DA). ROS generation was measured using flow cytometry. (**C**) Each bar represents the mean ± SD of three independent experiments. (**D**) After treatment with the indicated concentrations of isorhamnetin in the presence or absence of NAC, the content of intracellular ATP was measured. Each point represents the mean ± SD of three independent experiments (* *p* < 0.05, ** *p* < 0.001, and *** *p* < 0.0001 compared to control; ^##^
*p* <0.001 and ^###^
*p* < 0.0001 compared to isorhamnetin-treated cells). ISO—isorhamnetin.

**Figure 6 cancers-11-01494-f006:**
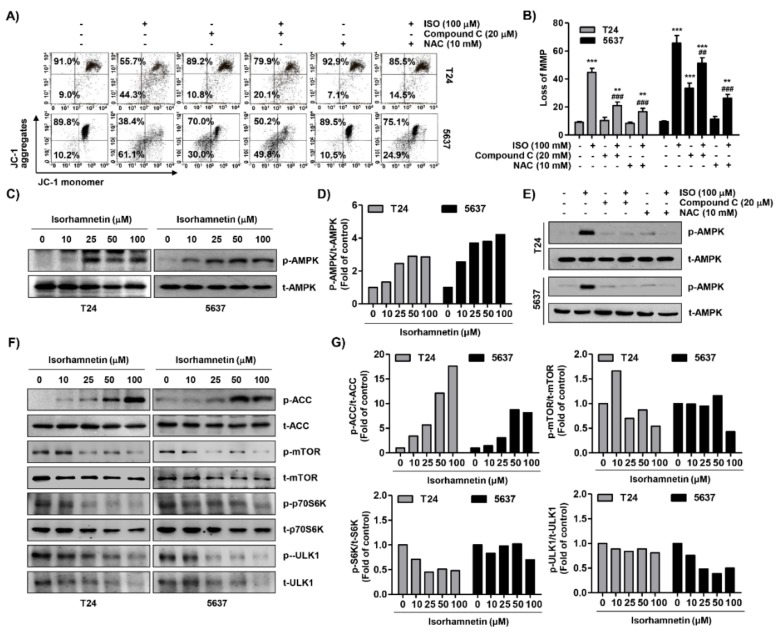
Mitochondrial dysfunction and activation of adenosine 5′-monophosphate-activated protein kinase (AMPK) signaling pathway using isorhamnetin in bladder cancer cells. (**A**,**B**) Cells were treated with 100 μM isorhamnetin for 48 h, or pre-treated with 20 μM compound C or 10 mM NAC for 1 h before isorhamnetin treatment for 48 h. (A) The cells were stained with 5,5′,6,6′-tetrachloro-1,1′,3,3′-tetraethylimidacarbocyanine iodide (JC-1) dye, and were then analyzed using flow cytometry in order to evaluate the changes in mitochondrial membrane potential (MMP). (**B**) Each bar represents the percentage of cells with JC-1 monomers (mean ± SD of triplicate determinations, ** *p* < 0.001 and *** *p* < 0.0001 compared to the control; ^##^
*p* < 0.001 and ^###^
*p* < 0.0001 compared to the isorhamnetin-treated cells). (**C**,**F**) After treatment with the indicated concentrations of isorhamnetin for 48 h, total cell lysates were prepared and Western blotting was then performed using the indicated antibodies and an ECL detection system. (**D**,**G**) The expression of each protein was indicated as a fold change relative to the control. Quantitative analysis of mean pixel density was performed using the ImageJ^®^ software. (**E**) The cells cultured under the same conditions as A and B were collected, and Western blotting was then performed.

**Figure 7 cancers-11-01494-f007:**
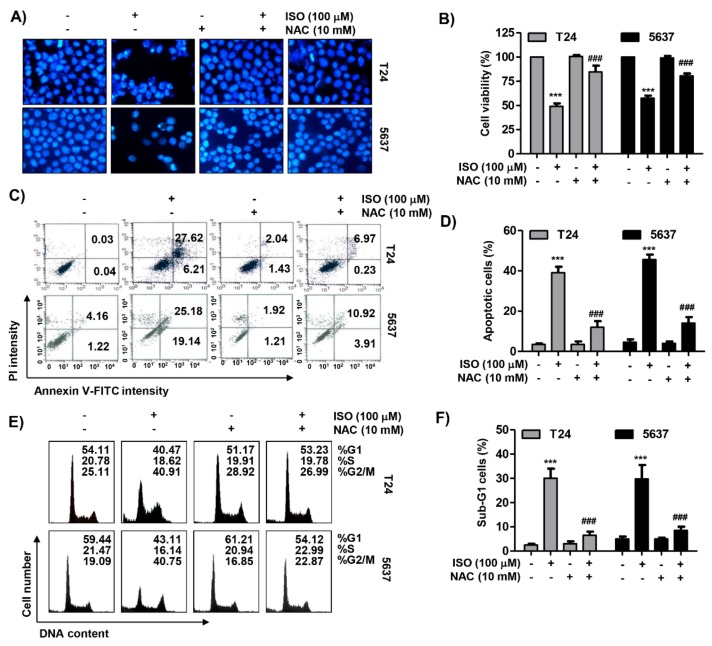
Roles of ROS in isorhamnetin-induced apoptosis and cell cycle arrest in bladder cancer cells. Cells were either treated with 100 μg/mL isorhamnetin for 48 h or pre-treated with 10 mM NAC for 1 h before isorhamnetin treatment, and were then collected. (**A**) The DAPI-stained nuclei were pictured under a fluorescence microscope. (**B**,**C**) The cells were stained with annexin V-FITC and PI for flow cytometry analysis. (B) Representative profiles. The results show early apoptosis, defined as annexin V^+^ and PI^−^ cells (lower right quadrant), and late apoptosis, defined as annexin V^+^ and PI^+^ (upper right quadrant) cells. (C) The percentages of apoptotic cells were determined by expressing the numbers of Annexin V^+^ cells as percentages of all the present cells. (**D**) The cells were stained with PI solution for flow cytometry analysis. Quantification of the cell population (in percent) in different cell cycle phases of viable cells is shown. (**E**) The percentages of apoptotic sub-G1 were calculated as the percentage of the number of cells in the sub-G1 population relative to the number of total cells. (**F**) The cell viability was assessed using an MTT assay. Each bar represents the mean ± SD of three independent experiments (*** *p* < 0.0001 compared to the control; ^###^
*p* < 0.0001 compared to the isorhamnetin-treated cells).

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
