# Peer review of "Isorhamnetin Induces Cell Cycle Arrest and Apoptosis Via Reactive Oxygen Species-Mediated AMP-Activated Protein Kinase Signaling Pathway Activation in Human Bladder Cancer Cells"

_cancers, 2019, doi:10.3390/cancers11101494_

Round 1

Reviewer 1 Report

Review of manuscript entitled: isorhamnetin Induces Cell Cycle Arrest at G2/M Phase and Apoptosis via ROS-Mediated AMPK Signaling Pathway Activation in Human Bladder Cancer Cells

Herein authors investigated the anti-cancer efficacy of isorhamnetin in human bladder cancer cells, focusing on the mechanisms associated with the induction of cell cycle arrest and apoptosis. Isorhamnetin triggers a G2/M cell cycle arrest, augments p21 expression and p21 bound to Cdk1 and increases both extrinsic and intrinsic apoptotic pathways. Moreover, isorhamnetin leads to a decrease in MMP, increased ROS production and decreased ATP production.  AMPK activation is downstream of the mitochondrial disfunction.

Manuscript is well written, easy to follow and conclusions are based on the results. Nevertheless, although results are interesting, there are major questions that should be addressed

.-Please, provide IC50 and LC50 (if reached) values for the effect on cell viability of isorhamnetin.

.-All images are of poor resolution and it is not easy to observe morphological changes (Fig. 1 B, Fig 7A), nor the FACS analysis.

.-Please, could you provide a quantification of the percentage of necrotic cells?

It has been described that energetic catastrophe (diminished ATP levels) can induce necrosis.

.-L152: Eliminate “apparently” as Fig 3B indicates increased p21 complexed to Cdk1.

.-L165: Modify “expression levels” to “protein levels.

.-L175: Modify “isorhamnetin-stimulated cells” to “ isorhamnetin-treated cells”.

.-It should be interesting to include in the discussion putative metabolic alterations associated to isorhamnetin. Is there any data about the effects of isorhamnetin on cell bioenergetics (aerobic glycolysis and mitochondrial oxidative phosphorylation)? Is there any result (or from bibliography) of the increase in autophagy as a putative complementary cause of cell death (autophagy induced cell death)?

Author Response

Herein authors investigated the anti-cancer efficacy of isorhamnetin in human bladder cancer cells, focusing on the mechanisms associated with the induction of cell cycle arrest and apoptosis. Isorhamnetin triggers a G2/M cell cycle arrest, augments p21 expression and p21 bound to Cdk1 and increases both extrinsic and intrinsic apoptotic pathways. Moreover, isorhamnetin leads to a decrease in MMP, increased ROS production and decreased ATP production. AMPK activation is downstream of the mitochondrial disfunction.

Manuscript is well written, easy to follow and conclusions are based on the results. Nevertheless, although results are interesting, there are major questions that should be addressed

.-Please, provide IC50 and LC50 (if reached) values for the effect on cell viability of isorhamnetin.

Answer) Thank you for your kind comment. We provide IC50 values of isorhamnetin on T24 cell and 5637 cells from Line 94 to Line 96, following as; “In addition, fifty percent inhibitory concentration (IC50) values of isorhamnetin on T24 and 5637 cells were 127.86 μM and 145.75 μM, respectively.”

.-All images are of poor resolution and it is not easy to observe morphological changes (Fig. 1 B, Fig 7A), nor the FACS analysis.

Answer) Thank you for your kind comment. All images including morphology and FACs analysis were replaced to easily observe.

.-Please, could you provide a quantification of the percentage of necrotic cells? It has been described that energetic catastrophe (diminished ATP levels) can induce necrosis.

Answer) Thank you for your kind comment. We proved the percentage of necrotic cells upon isorhamnetin-treated T24 and 5637 cells in Figure 2B. In addition, we added the results at Line 130-131.

.-L152: Eliminate “apparently” as Fig 3B indicates increased p21 complexed to Cdk1.

Answer) Thank you for your kind comment. We deleted the word.

.-L165: Modify “expression levels” to “protein levels.

Answer) Thank you for your kind comment. We changed the word.

.-L175: Modify “isorhamnetin-stimulated cells” to “ isorhamnetin-treated cells”.

Answer) Thank you for your kind comment. We changed the word.

.-It should be interesting to include in the discussion putative metabolic alterations associated to isorhamnetin. Is there any data about the effects of isorhamnetin on cell bioenergetics (aerobic glycolysis and mitochondrial oxidative phosphorylation)? Is there any result (or from bibliography) of the increase in autophagy as a putative complementary cause of cell death (autophagy induced cell death)?

Answer) Thank you for your important comment. First, we have performed additional experiments for identification of effect of isorhamnetin on downstream of AMPK including phosphorylation of mTOR and ULK1 which regulate cell proliferation, apoptosis and autophagy. Our data shown that exposure of T24 and 5637 cells with isorhamnetin led to down-regulates in phospholyation of mTOR, p76S6K and ULK1 in a dose-dependent manner (Figures 6F and G). We added this findings in “Results” and “Discussion” parts. Next, we evaluated the effects of isorhamnetin on expression of autophagy-related factors in T24 cells. A few studies reported that isorhamnetin induced the expression of LC3II, beclin1, p62 and ATG in human breast cancer cells and non-small cell lung cancer cells, thereby mediated autophagic cell death (Mol Med Rep. 2015 Oct;12(4):5796-806.; Sci Rep. 2018 Jul 26;8(1):11255.). Interestingly, our finding obtained from T24 bladder cancer cells had offered conflicting previous reported. Our results showed that the expression of these autophagy-related markers was markedly decreased in isorhamnetin-treated T24 bladder cancer cells.

Therefore, we considered that further studies are required for identification of why isorhamnetin down-regulated the expression of autophagy markers in bladder cancer cells, and whether this situation occur only human bladder cancer T24 cells. In addition, although we can not identified the effects of isorhamnetin on cell bioenergetics in T24 and 5637 cells, we explained the significance of that in “Discussion” part. Moreover, we also elucidated the need to the further studies for identification of isorhamnetin-mediated AMPK signaling on mitochondrial energy metabolism.

Reviewer 2 Report

Reviewer’s report for Cancers-557848

In the article entitled "Isorhamnetin Induces Cell Cycle Arrest at G2/M Phase and Apoptosis via ROS-Mediated AMPK Signaling Pathway Activation in Human Bladder Cancer Cells", Park et al have studied the effects of isorhamnetin on cell cycle progression, survival, mitochondrial function, and related gene expression in bladder cancer cell lines T24, 5637, 2531J and EJ. They showed that isorhamnetin was able to induce cell cycle block at G2/M, due to a decrease in Wee1 and cyclin B1 expression, with upregulation of p21, and display a negative effect on survival through apoptosis. Isorhamnetin-induced G2/M arrest and apoptosis were proved to be mediated by ROS-dependent mitochondrial dysfunction associated with disturbance of ATP production, which resulted in activation of the AMPK signaling pathway in bladder cancer cells.

Research on the effects of isorhamnetin on cells goes back at least to 1996, when phenolics in the human diet (among which isorhamnetin) were shown to affect pharmacokinetics of carcinogens by (i) altering the rates of their absorption and uptake, (ii) reacting with or tightly binding to toxic substances and metabolites, and (iii) modifying cytochrome P450-dependent metabolism and phase II enzymes [Malaveille et al (1996) Carcinogenesis vol.17 no.10, pp.2193-2200; Malaveille et al (1998) Mutation Research 402, 219–224]. From then on, numerous studies have been performed dealing with the anti-cancer effects of isorhamnetin, but also proving that these were accompanied by disturbances in a variety of signaling pathways [Ramachandran L et al (2012) JBC vol. 287, no. 45, pp. 38028–38040; Saud S et al (2013) Cancer Research 73(17), 5473–84; Kim J-E et al (2011) Cancer Prevention Research 4(4), 582–91; Li C et al (2014) Molecular Medicine Reports 9, 935-940; Zhang H-W et al (2018) Scientific Reports 8, 11255; Manu KA et al (2015) Cancer Letters 363, 28–36; Qi F et al (2018) Microbial Pathogenesis 120, 37–41; Hu J et al (2019) Journal of Experimental & Clinical Cancer Research 38, 225]. Based on the above, the ROS-mediated anti-proliferative and pro-apoptotic properties of isorhamnetin seem to be well-established. However, the present paper constitutes the first research work on urothelial bladder cancer cells.

There are no reservations for the scientific approach or the tools used and procedures followed in this work. But although the real question here would be by which mechanisms isorhamnetin induces mitochondrial dysfunction to stimulate the production of intracellular ROS, and not simply whether isorhamnetin harbors anti-proliferative and pro-apoptotic properties, nevertheless, the paper is decently structured and the information therein could be of interest to scientists activated in the field. So, it may be published in Cancers after a few major issues are solved.

Major points

The findings of this paper show that ROS induces cell cycle arrest, apoptosis, and AMPK activation. So, the title of the paper (Isorhamnetin Induces Cell Cycle Arrest at G2/M Phase and Apoptosis via ROS-Mediated AMPK Signaling Pathway Activation in Human Bladder Cancer Cells) which implies that AMPK pathway activation results in cell cycle arrest and apoptosis is erroneous. Please change it to something like “Isorhamnetin Induces ROS-Mediated Cell Cycle Arrest at G2/M Phase, Apoptosis, and AMPK Signaling Pathway Activation in Human Bladder Cancer Cells”. Cell line named EJ in this paper (which in reality must be EJ-1) has been shown to actually be T24. Please have a look at the cellosaurus links below,

EJ: https://web.expasy.org/cellosaurus/CVCL_7039

EJ-1: https://web.expasy.org/cellosaurus/CVCL_2893

and the following abstract from the paper by O’Toole CM et al (1983) Nature 301, 429-430, doi:10.1038/301429a0.

Then, make the corrections needed by eliminating EJ-1 cells from this work.

Abstract

Recent reports on transfection of mouse cells with DNA from the established human urinary bladder cancer cell lines T24, J82 and EJ (MGH-U1), and the presence of an identical genetic modification in T24 and EJ cells have led us to examine the identity of these and other cultures of urothelial origin. By the criteria of HLA-A-B-C typing and isozyme analysis, we conclude that EJ (MGH-U1) and some cultures of J82 are in fact T24 cells. However, five other bladder cancer cell lines, J82 (COT), RT4, RT112, TCCSuP and SCaBER, are clearly distinct from T24 by HLA typing and/or isozyme patterns.

The experimental evidence on isorhamnetin anti-cancer effects provided here relates almost entirely to the effects attributed to ROS generation. Because data on the involvement of AMPK pathway is missing from the relevant literature, the paper is going to gain in strength if the authors go on and study some AMPK pathway downstream effectors like mTOR (S6K phosphorylation), ULK1 (phosphorylation and aurophagy detection), ATGL (phosphorylation), or others, by implementing the additional experiments needed. In concert with a necessary presentation of the AMPK pathway and effects resulting during its activation, a brief discussion on signaling pathways known to be affected by isorhamnetin should be added in the discussion section. (A) The mechanism of action of isorhamnetin involves a marked increase in ROS levels (this paper and others);

(B) High levels of ROS are shown to activate the PI3K-Akt pathway [Zhang J et al (2016) Oxidative Medicine and Cellular Longevity, vol 2016, Article ID 4350965, http://dx.doi.org/10.1155/2016/4350965, and Koundouros N and Poulogiannis G (2018) Frontiers in Oncology 8, 160, doi: 10.3389/fonc.2018.00160];

If (A) and (B) are true, then how can it be that cancer cells with an activated PI3K-Akt pathway (due to high ROS levels) may be blocked at G2/M and undergo apoptosis when challenged with isorhamnetin? Please comment briefly. (Remember that T24 cells bear an activated H-Ras [Homozygous for H-Ras p.Gly12Val (c.35G>T) (PubMed=12068308; ATCC)], whereas 5637 are wild type for H-Ras.)

Author Response

Major points

The findings of this paper show that ROS induces cell cycle arrest, apoptosis, and AMPK activation. So, the title of the paper (Isorhamnetin Induces Cell Cycle Arrest at G2/M Phase and Apoptosis via ROS-Mediated AMPK Signaling Pathway Activation in Human Bladder Cancer Cells) which implies that AMPK pathway activation results in cell cycle arrest and apoptosis is erroneous. Please change it to something like “Isorhamnetin Induces ROS-Mediated Cell Cycle Arrest at G2/M Phase, Apoptosis, and AMPK Signaling Pathway Activation in Human Bladder Cancer Cells”. Cell line named EJ in this paper (which in reality must be EJ-1) has been shown to actually be T24. Please have a look at the cellosaurus links below,

EJ: https://web.expasy.org/cellosaurus/CVCL_7039

EJ-1: https://web.expasy.org/cellosaurus/CVCL_2893

and the following abstract from the paper by O’Toole CM et al (1983) Nature 301, 429-430, doi:10.1038/301429a0.

Then, make the corrections needed by eliminating EJ-1 cells from this work.

Answer) Thank you for your kind comment. We eliminated the results obtained from EJ-1 cells.

Abstract

Recent reports on transfection of mouse cells with DNA from the established human urinary bladder cancer cell lines T24, J82 and EJ (MGH-U1), and the presence of an identical genetic modification in T24 and EJ cells have led us to examine the identity of these and other cultures of urothelial origin. By the criteria of HLA-A-B-C typing and isozyme analysis, we conclude that EJ (MGH-U1) and some cultures of J82 are in fact T24 cells. However, five other bladder cancer cell lines, J82 (COT), RT4, RT112, TCCSuP and SCaBER, are clearly distinct from T24 by HLA typing and/or isozyme patterns.

The experimental evidence on isorhamnetin anti-cancer effects provided here relates almost entirely to the effects attributed to ROS generation. Because data on the involvement of AMPK pathway is missing from the relevant literature, the paper is going to gain in strength if the authors go on and study some AMPK pathway downstream effectors like mTOR (S6K phosphorylation), ULK1 (phosphorylation and aurophagy detection), ATGL (phosphorylation), or others, by implementing the additional experiments needed. In concert with a necessary presentation of the AMPK pathway and effects resulting during its activation, a brief discussion on signaling pathways known to be affected by isorhamnetin should be added in the discussion section.

Answer) Thank you for your kind comment. We have performed additional experiments for identification of effect of isorhamnetin on downstream of AMPK including phosphorylation of mTOR and ULK1 which regulate cell proliferation, apoptosis and autophagy. Our data shown that exposure of T24 and 5637 cells with isorhamnetin led to down-regulates in phospholyation of mTOR, p76S6K and ULK1 in a dose-dependent manner (Figures 6F and G). We added this findings in “Results” and “Discussion” parts.

(A) The mechanism of action of isorhamnetin involves a marked increase in ROS levels (this paper and others); (B) High levels of ROS are shown to activate the PI3K-Akt pathway [Zhang J et al (2016) Oxidative Medicine and Cellular Longevity, vol 2016, Article ID 4350965, http://dx.doi.org/10.1155/2016/4350965, and Koundouros N and Poulogiannis G (2018) Frontiers in Oncology 8, 160, doi: 10.3389/fonc.2018.00160];

If (A) and (B) are true, then how can it be that cancer cells with an activated PI3K-Akt pathway (due to high ROS levels) may be blocked at G2/M and undergo apoptosis when challenged with isorhamnetin? Please comment briefly. (Remember that T24 cells bear an activated H-Ras [Homozygous for H-Ras p.Gly12Val (c.35G>T) (PubMed=12068308; ATCC)], whereas 5637 are wild type for H-Ras.)

Answer) Thank you for your important comment. Urothelial bladder cancer is typically a disease of adults in the middle to late decades of life characterized by mutations in tumor suppressor genes such as p53, RB, and PTEN, which have been found to be independent predictors for progression [J Urol. 2007;177:481-7]. Beukers et al reported that mutation in codon 12 (glycine to valine) of H-RAS was consistently detected in young age [Eur J Hum Genet. 2014;22:837-9]. H-RAS was mutated with a higher frequency in tumors of patients younger than 20 years than in adults. Furthermore, mosaicism for oncogenic H-RAS mutations may increase the risk for developing bladder cancer at a very young age. Contrary to pediatric cases, mutation of H-RAS is detected only in 5% of bladder urothelial cancer in adults [PloS ONE 2010;5:e13821]. This compelling differences support our hypothesis that pediatric urothelial bladder tumors may evolve through distinct molecular alterations centered on the H-RAS gene. In present study, we used T24 cells and 5637 cells purchased from ATCC. T24 cells origin from 81 age adult (ATCC® HTB-4TM) and 5637 cells origin from 68 years adult (ATCC® HTB-9TM). Therefore, our data shown that isorhamnetin-mediated accumulated ROS induced cell cycle arrest and apoptosis in bladder cancer cells that origin from aged adult, regardless of mutation of H-RAS gene.

Round 2

Reviewer 1 Report

I think that authors have answered my comments,
and the manuscript should be suitable for publication. 

Author Response

Answer) Sincere thanks for your decision.

Reviewer 2 Report

Reviewer’s report for Cancers-557848 revised version

In its revised form, the article entitled "Isorhamnetin Induces Cell Cycle Arrest at G2/M Phase and Apoptosis via ROS-Mediated AMPK Signaling Pathway Activation in Human Bladder Cancer Cells", by Park et al is acceptable for publication in Cancers, with the following two reservations:

1)One of the main points of this paper is (I copy from the title) ROS-Mediated AMPK Signaling Pathway Activation. In fact, (a) in the Discussion, although the authors discuss extensively the issue of ROS-mediated cell cycle block and apoptosis, they write too little on the possible involvement of AMPK activation and the possible cross-talk between AMPK and pathways associated to cell proliferation, autophagy and/or apoptosis (which is far less straightforward); and (b) the explanation they provide (lines 344-348) when discussing the results of Fig 6F is hasty and not clear. The western blot for non-phosphorylated ULK1 in both cell lines (T24 and 5637 – please add the name of the cell lines below Fig 6F) shows a clear downregulation with growing concentrations of isorhamnetin. Therefore, contrary to the results from MDA-MB-231 cells, this is likely to mean that isorhamnetin is not inducing autophagy in T24 and 5637 cells, but, as the authors write in line 347, isorhamnetin may induce only apoptosis and not autophagy in these bladder cancer cells.

2)In the revised areas of the new manuscript, there is need for editing, due to several language errors.

Author Response

1)One of the main points of this paper is (I copy from the title) ROS-Mediated AMPK Signaling Pathway Activation. In fact, (a) in the Discussion, although the authors discuss extensively the issue of ROS-mediated cell cycle block and apoptosis, they write too little on the possible involvement of AMPK activation and the possible cross-talk between AMPK and pathways associated to cell proliferation, autophagy and/or apoptosis (which is far less straightforward); and (b) the explanation they provide (lines 344-348) when discussing the results of Fig 6F is hasty and not clear. The western blot for non-phosphorylated ULK1 in both cell lines (T24 and 5637 – please add the name of the cell lines below Fig 6F) shows a clear downregulation with growing concentrations of isorhamnetin. Therefore, contrary to the results from MDA-MB-231 cells, this is likely to mean that isorhamnetin is not inducing autophagy in T24 and 5637 cells, but, as the authors write in line 347, isorhamnetin may induce only apoptosis and not autophagy in these bladder cancer cells.

Answer) Thanks for your kind comments. We explained the significance of ROS-mediated AMPK activation and AMPK-associated cell fate in Discussion part. Moreover, we added name of the cell lines below Figure 6F, and revised discussion of the result from Figure 6F.

Based on Figure 6F, we conjecture that AMPK-mediated interruption of mTOR/p70S6K/ACC1 signaling pathway may contribute to isorhamnetin-induced cell cycle arrest and apoptosis. However, we found that isorhamnetin down-regulated the expression and phosphorylation of ULK1, a trigger of autophagy. Therefore, we have performed additional experiments for identification of the effect of isorhamnetin on autophagy in T24 cells. Our supplementary result was showed that the expression of autophagy-related markers was down-regulated by isorhamnetin treatment in common with ULK1. Therefore, our data suggested that isorhamnetin-induced ROS activates AMPK, and subsequently down-regulate the mTOR/ACC1/ULK1 signaling pathway which result in promotes cell apoptosis and inhibit autophagy at the same time, it due to at the same time. However, we considered that further studies are required for identification of why isorhamnetin down-regulated the expression of autophagy markers in bladder cancer cells, and whether this situation occur only human bladder cancer T24 cells.

2)In the revised areas of the new manuscript, there is need for editing, due to several language errors.

Answer) Thank you for reviewer’s comment. Our manuscript has been edited by an English language editing service.
